# The Essential Oil of *Cymbopogon citratus* Stapt and Carvacrol: An Approach of the Antitumor Effect on 7,12-Dimethylbenz-[α]-anthracene (DMBA)-Induced Breast Cancer in Female Rats

**DOI:** 10.3390/molecules25143284

**Published:** 2020-07-20

**Authors:** Juan Pedro Rojas-Armas, Jorge Luis Arroyo-Acevedo, Miriam Palomino-Pacheco, Oscar Herrera-Calderón, José Manuel Ortiz-Sánchez, Agustín Rojas-Armas, James Calva, Américo Castro-Luna, Julio Hilario-Vargas

**Affiliations:** 1Laboratory of Pharmacology, Faculty of Medicine, Universidad Nacional Mayor de San Marcos, Av. Miguel Grau 755, Lima 15001, Peru; jprojasarmas@yahoo.com (J.P.R.-A.); jlarroyoa@gmail.com (J.L.A.-A.); 2Laboratory of Biochemistry, Faculty of Medicine, Universidad Nacional Mayor de San Marcos, Av. Miguel Grau 755, Lima 15001, Peru; mirianpp7@hotmail.com; 3Department of Pharmacology, Bromatology and Toxicology, Faculty of Pharmacy and Biochemistry, Universidad Nacional Mayor de San Marcos, Jr. Puno 1002, Lima 15001, Peru; 4Laboratory of Physiology, Faculty of Medicine, Universidad Nacional Mayor de San Marcos, Av. Miguel Grau 755, Lima 15001, Peru; josemanuel4470@yahoo.es; 5Instituto Regional de Enfermedades Neoplásicas, Trujillo 13007, Peru; agustin_rojasa@hotmail.com; 6Departamento de Química y Ciencias Exactas, Universidad Técnica Particular de Loja, San Cayetano Alto s/n, Loja 1101608, Ecuador; jwcalva@utpl.edu.ec; 7Research Institute for Pharmaceutical Sciences and Natural Resources, Faculty of Pharmacy and Biochemistry, Universidad Nacional Mayor de San Marcos, Jr. Puno 1002, Lima 15001, Peru; acastrol@unmsm.edu.pe; 8Department of Physiology, Faculty of Medicine, Universidad Nacional de Trujillo, Trujillo 13007, Peru; jhilario@unitru.edu.pe

**Keywords:** *Cymbopogon citratus*, cancer experimental, breast cancer, essential oil, carvacrol, carcinogenic agent

## Abstract

*C. citratus* essential oil and carvacrol have shown an antitumor effect on breast tumor cell lines; the main objective of this research was to evaluate the antitumor effect of the essential oil of *Cymbopogon citratus* (EOCc) and carvacrol on 7,12-dimethylbenz [a] anthracene (DMBA)-induced breast cancer in female rats. Cancer was induced by a single administration of DMBA at dose of 80 mg/kg body weight (BW). A total of 54 female Holtzman rats were randomly assigned into 9 groups (n = 6). Group I: PS (Physiological saline); Group II: DMBA; Groups III, IV, and V: DMBA + EOCc at doses of 50, 100 and 200 mg/kg/day BW, respectively; Groups VI, VII, and VIII: DMBA + carvacrol at doses of 50, 100 and 200 mg/kg/day BW, respectively; and group IX: DMBA + EOCc + carvacrol at doses of 100 mg/kg/day BW. The treatment lasted 14 weeks. As results, EOCc showed a reduction in tumors as well as necrosis and mitosis. Animals treated with carvacrol did not show necrosis, mitosis, or infiltration. Carvacrol at dose of 100 mg/kg/day BW revealed a significant decrease in the cumulative tumor volume down to 0.11 ± 0.05 cm^3^ compared to 0.38 ± 0.04 cm^3^ of the DMBA group (*p* < 0.01). It is concluded that EOCc and carvacrol had an antitumor effect on DMBA-induced breast cancer in female rats.

## 1. Introduction

Breast cancer is the most common cancer among women worldwide, and accounts for 16% of all female cancers; however, the incidence varies widely across different parts of the world, with age-standardized rates of up to 99.4 per 100,000 in North America, while in Eastern Europe, South America, South Africa, and Western Asia have moderate incidents, but are increasing yearly [1]. This type of cancer is the most common in Latin America and the Caribbean women, both in the number of new cases and deaths; in 2012, more than 408,000 women were diagnosed with breast cancer in this region, and 92,000 have died from this disease. Projections indicate that the number of women diagnosed with breast cancer in the Americas will increase by 46% in 2030. In 2012, more than 408,000 women were diagnosed with breast cancer in this region, and 92,000 died from this disease. Projections indicate that the number of women diagnosed with breast cancer in the Americas will increase by 46% in 2030 [2].

DMBA (7,12-dimethylbenz-[α]-anthracene) is a commercial chemical agent used to induce breast cancer in preclinical models by activating cellular cytosolic receptor aryl hydrocarbon receptor (AhR). When AhR is activated, this leads to a translocation into the nucleus and combines with AhR nuclear translocation protein. The AhR/ARNT complex induces gene transcription by binding to specific DNA recognition sites upstream to AhR responsive genes. AhR-dependent up-regulation of cytochrome P450 (CYP1A1 and CYP1B1) enzymes is involved in metabolizing DMBA into a mutagenic epoxide intermediate that forms DNA adducts associated with polycyclic aromatic hydrocarbons (PAH)-mediated carcinogenesis [3].

There are several kinds of drugs for breast cancer treatment that are used according to the characteristics of the tumor and the disease scale. Drugs can be used in systemic chemotherapy, endocrine therapy, or HER2 (human epidermal growth factor receptor 2)-directed therapy. However, there is concern regarding an increased risk of side effects from these drugs [4]. On the other hand, the cost of drugs for breast cancer treatment is very high [5]. This situation motivates the search for new agents with a better safety profile and lower cost. In this context, an alternative source is natural resources based on phytotherapy. Medicinal plants are used by traditional medicine to treat many diseases, including cancer. A substantial number of cancer patients resort to herbal remedies for complementary treatment [6]. It has been reported that a large number of phytochemicals have anti-cancer properties, including polyphenols that have anti-radical activity and work as antioxidants [7]. However, herbal medicines may present a potential risk of adverse effects and interactions due to herbal drugs, usually a mixture of various bioactive compounds, which can interfere with different metabolic pathways and can modify the action of physical therapies [8,9]. Moreover, the activity of herbal drugs can be diversified in physiological and pathological conditions.

*Cymbopogon citratus* Stapt (*C. citratus*) is a frost-tender clumping perennial grass of the *Poaceae* family (*Gramineae*), native to Southeast Asia, but currently grows around the world, mainly in tropical regions and savannas; known by the common name, according to the different countries, such as: “hierba luisa”, “caña Santa”, “lemon-grass”, “te lemon”, “citronella”, etc. *C. citratus* forms a dense rounded clump of foliage to 0.60–0.90 m (less frequently to 1.20 m) tall and as wide in one growing season. Gracefully arching, strap-shaped linear leaves (to 0.90 m long and 0.02 m wide) are light green. Leaves emit a lemony fragrance when bruised. It rarely produces flowers [10].

Pharmacological studies have demonstrated several properties of the essential oil of *Cymbopogon citratus* (EOCc), such as: antifungal [11,12]; antibacterial [13,14]; anxiolytic, sedative, anticonvulsant [15,16,17], anti-*Trypanosoma cruzi* [18]; hypotension associated with bradycardia in non-anesthetized normotensive rats and vasodilator in rat mesenteric artery [19]; also, the main constituent of the EOCc, citral, showed significant anti-inflammatory effect due to the inhibition of nitric oxide production through the suppression of NFkB activity and, on lipopolysaccharide activated cells induced hypoexpression of COX-2 (Cyclooxygenase 2) and TNF-α (Tumor necrosis factor-α) [18]. On the other hand, main phytochemical components found in EOCc were neral, citral or geranial, camphene, nonan-4-ol, 6-metil-hept-5-en-2-one, and citronelal. Furthermore, these chemical constituents could vary depending on the extracted organ, plant age, geographical area of the collection and the extraction method to obtain the essential oil [20].

Regarding its antineoplastic potential, the EOCc showed an anti-cancer effect on cervical cancer cell lines and breast cancer cell line, MCF-7 [21,22,23] Likewise, citral induced apoptosis in several hematopoietic cancer cell lines, accompanied by DNA fragmentation and induction of the catalytic activity of caspase-3 [24], and in ovarian cancer cell lines [25,26]. Additionally, carvacrol, a monoterpenes-phenol is found in the essential oil of many aromatic plants of the Lamiaceae family, including thyme and oregano, showed its anti-cancer potential on the human metastatic breast cancer cell line [27].

The effectiveness of chemopreventing agents reflects their ability to counteract certain biochemistry signals linked to carcinogenesis such as genotoxic damage, redox imbalances, and other forms of cellular stress. Chemoprevention by edible phytochemicals is now considered to be an inexpensive, readily applicable, acceptable and accessible approach to cancer control and management [28]. Antitumor activity of natural products should have: (1) high efficacy in multiple sites; (2) capability of oral consumption; (3) little or no toxicity; (4) known mechanisms of action; (5) low cost, and human acceptance [28]. Phytochemicals in essential oil have been studied focusing the mechanistic insight into chemoprevention further includes several targets such as Nf-kB, AP-1, β-catenin, proinflammatory mediators, proinflammatory enzymes COX-2/iNOS and protein kinases [29]. However, all these studies have been carried out with tumor cell lines, not founding any result with in vivo study, so we proposed as main objectives as follows: (1) to evaluate the antitumor effect of the essential oil of *Cymbopogon citratus* and carvacrol on DMBA induced breast cancer in female rats; (2) to identify the chemical constituents of EOCc; (3) to evaluate the antioxidant activity of EOCc and carvacrol against DPPH assay; and (4) to evaluate any histopathological changes of mammary tissues in order to observe any protective effect of the two evaluated substances as well as any synergic effect.

## 2. Results

### 2.1. Identification of the Chemical Constituents of the Essential Oil from C. citratus

A total of 22 compounds were identified in the EOCc (Table 1). The most abundant components were geranial (40.45%) and neral (31.84%); these are optical isomers: geranial, trans (citral A) and neral, cis (citral B) that form citral, so that the citral represented 72.29% of the total composition. Myrcene represented 13.60%. The chromatogram is shown in Figure 1.

### 2.2. Antioxidant Activity of EOCc and Carvacrol by Using the DPPH Method

Figure 2 shows that EOCc and carvacrol showed antioxidant activity by the DPPH method, with values of IC_50_ of 48 mg/mL and 145 µg/mL, respectively.

### 2.3. Antitumor Effect of the Essential Oil from C. citratus and Carvacrol on DMBA-Induced Breast Cancer in Rats

#### 2.3.1. The Cumulative Tumor Volume in Animals Treated with EOCc and Carvacrol

Table 2 shows the results of treatment for 14 weeks with EOCc and carvacrol on DMBA-induced breast cancer in rats. It was observed that 100% of the animals in the group that received DMBA developed carcinogenesis, with the highest values in number of tumors per group, lower tumor latency, and greater cumulative tumor volume. With the EOCc at doses of 50, 100 and 200 mg/kg/day, there was a discreetly dose-dependent effect, thus reducing the total number of tumors, decreasing the frequency of tumors per group (44% less at dose of 200 mg/kg/day BW), decrease in the average tumor volume and cumulative tumor volume. The best effect was observed with carvacrol at the dose of 100 mg/kg/day, with a total number of four tumors compared to 16 tumors of the DMBA group, a 75% reduction in the frequency of tumors per group, reduction of 67% of the incidence of tumors, increased tumor latency up to 79.00 ± 7.00 days compared to 65.17 ± 2.78 days of the DMBA group, a significant decrease in the average tumor volume to 0.11 ± 0.05 cm^3^ compared to 0.38 ± 0.04 cm^3^ of the group DMBA (*p* < 0.001), and reduction to 0.44 cm^3^ of cumulative tumor volume compared with 6.10 cm^3^ of the DMBA group. With the treatment of EOCc and carvacrol at doses of 100 mg/kg/day BW, a 56% reduction in the frequency of tumors was observed, increased tumor latency to 77.25 ± 3.66 days compared to 65.17 ± 2.78 days of DMBA group, reduction of average tumor volume and cumulative tumor volume (Figure 3 and Table 2).

#### 2.3.2. The Body Weight Variations of Animals Treated with EOCc and Carvacrol

The body weight of the animals was controlled every week (Figure 4). A significant decrease in body weight gain (*p* < 0.05) was observed in the DMBA group, starting at the eighth week (210 ± 12 g); the DMBA + EOCc 100 mg/kg/day (220 ± 10 g) and DMBA + carvacrol 200 mg/kg/day (218 ± 9 g) groups compared to control group. The rats’ body weight decreased significantly in the DMBA group with the average value of 223.67 ± 9.61 g compared to control group 246.33 ± 8.31 g (*p* < 0.05) at the end of the experiment.

#### 2.3.3. Histopathological Findings in the Experimental Breast Cancer

Histopathological analysis showed extensive tumor necrosis that reached 75% in the DMBA group (Table 3, Figure 5B). Treatment with the EOCc decreased the percentage of necrosis and mitosis (Figure 5C–E). With carvacrol, no tumor necrosis was observed (Figure 5F,G). The best effect was observed with carvacrol at a dose of 100 mg/kg/day, where no necrosis, mitosis, or infiltration was observed. With the combination of EOCc and carvacrol at doses of 100 mg/kg/day each, there was an absence of necrosis and a decrease in mitosis (Table 3 and Figure 5H).

## 3. Discussion

Citral (geranial and neral) was the most abundant component in the EOCc (72.29%), followed by myrcene (13.60%). This result is consistent with other studies that also reported citral as the main component, although with slight variations in the percentage; thus in Brazil (samples from three different places) 69.31% [31], 75.27% and 89.57% were reported [32]; Cuba, 86.35% [33]; West Africa, 82.55%; [34] Senegal: Dakar, 77.8% and Kaolack, 82.8%; Egypt, 79.69%; Saudi Arabia, 71.4% and India (Bangalore), 72.2%. Myrcene was found in amounts close to our study, in Brazil, 17.58% and Senegal, 10.8% [11,31,32,33,34,35]. These variations could be linked to the place, season, and harvesting techniques where *Cymbopogon citratus* was collected as well as the methodology or equipment used to obtain the essential oil.

The antioxidant activity against DPPH of the EOCc was slight (IC_50_ = 48 mg/mL), which is close to the result of the study of a sample from Minas Gerais (Brazil) that showed a minor inhibitory effect in DPPH radical assay [36]. However, various results have been reported with samples from other places; such as India, 69% of inhibition at 0.1 mg/mL of concentration; West Africa, 67.6% at 8 mg/mL; Egypt, IC_50_ = 1 mg/mL, and Saudi Arabia, IC_50_ = 6.9 mg/mL [37]. On the other hand, in this study, carvacrol showed antioxidant activity with an IC_50_ = 145 µg/mL (Figure 2). Another study that used the same method reported an IC_50_ = 12.03 µg/mL for carvacrol [38].

The EOCc at the evaluated doses produced a reduction in the total number of tumors, a decrease in the frequency of tumors per group (44% less with the dose of 200 mg/kg/day), as well as in the average tumor volume and the cumulative tumor volume (Table 2, Figure 3); also, at the histological level, a decrease in the percentage of necrosis and mitosis was observed (Figure 5C–E). An in vitro study showed cytotoxicity of the EOCc on MCF-7 breast cancer cells [22,39]. It was also effective on LNCaP and PC-3 prostate cell lines, and on SF-767 glioblastoma cell lines, showing that the activity of the EOCc was statistically equal to its main component, citral [23]. Therefore, it is likely that the anti-tumor effect of the EOCc observed in our study could be due to the high content of citral in the sample. A study reported with EOCc and citral produced a decreased expression of Bcl-xL and Mcl-1 antiapoptotic factors, in turn, suppressed the proliferation/survival of small lung cancer cells (SCLC cells), [23,39] also decreased cell proliferation, increasing intracellular (reactive oxygen species) ROS, altering the potential of mitochondrial membrane, and initiating apoptosis in cervical cancer cell lines HeLa and ME-180 [40]. However, we did not test additional assays on breast cancer cell line to stablish any relationship between the antioxidant activity and breast cancer progression.

The citral has shown an antitumor effect on breast cancer cell lines such as MDA-MB-231 and MCF-7, in this case with G2/M phase cycle arrest and apoptosis induction [26]; likewise, citral was an effective aldehyde dehydrogenase 1A3 (ALDH1A3) inhibitor and capable of blocking the growth of the ALDH1A3-mediated breast tumor, potentially blocking its colony formation and gene expression regulation activity [41], but it also inhibited the growth of 4T1 breast tumors in Balb/c mice, although geranial (citral A) had a significant effect on mice induced with breast cancer 4T1 cell line [42]. In addition, the citral showed cytotoxic activity on other cancer types, such as the HepG2 cell line, human colorectal cancer cells HCT116 and HT29, where it induced mitochondrial-mediated apoptosis via increased intracellular ROS, induced phosphorylation of the p53 protein and the Bax expression while decreasing the expression of Bcl-2 and Bcl-xL, which promoted the cleavage of caspase-3 [43].

In the present investigation, the best results were obtained with carvacrol in doses of 100 mg/kg/day, followed by the combination of EOCc and carvacrol in doses of 100 mg/kg/day (Table 2 and Table 3), which was linked to histological findings (Figure 5). Although this study does not present any result on apoptotic factors, in other studies, carvacrol has demonstrated an antiproliferative, anti-metastatic, and anti-angiogenic effect on human metastatic breast cancer cells MDA-MB-231 [27]. Carvacrol also had a beneficial effect against 1,2-dimethylhydrazine-induced colon carcinogenesis in rats; cytotoxicity, apoptosis and DNA damage in AGS human gastric adenocarcinoma cells; and liver cancer [44] by inhibiting cell proliferation and prevented metastasis in hepatocellular carcinogenesis induced by diethylnitrosamine in rats [44]; significantly inhibited tumor cell proliferation, metastasis and invasion; and also induced apoptosis in human oral squamous cell carcinoma (OSCC) with G1/S cell cycle arrest, Bcl-2 downregulation, Cox-2 and Bax upregulation; induced apoptosis in A549 human pulmonary adenocarcinoma cells by ROS induction; presented strong antitumor potential in vivo using an athymic mouse model; decreased the rate of cell proliferation in N2a neuroblastoma cells; blocked TRPM7 type currents in PC-3 and DU-145 prostate cancer cells; and reduced their proliferation, migration, and invasion [45,46].

The decrease in the body weight of experimental animals is an important indicator of toxicity. In the present study, the weekly control of the rats’ body weight revealed that the EOCc did not significantly affect weight gain, which would indicate an apparent lack of toxicity [47]. However, in addition to the DMBA group, in the groups treated with carvacrol 100 mg/kg/day and EOCc 100 mg/kg/day + carvacrol 100 mg/kg/day a significant decrease (*p* < 0.05) of body weight was observed (Figure 4). Information on the toxicology of carvacrol is limited, and there are no data on its oral toxicity in repeated doses. Since the best effect against experimental breast cancer was observed with these two groups that received carvacrol, it is advisable to carry out a subacute and subchronic oral toxicity tests to establish the safety level of carvacrol. Additionally, the EOCc resulted safety at doses of 2000 mg/kg by oral administration according to Lulekal et al., 2019 [48].

It is likely that the antitumor activity demonstrated in this study due to the effect of carvacrol treatment may be due to its antioxidant activity (Figure 2). It is important to highlight the role of reactive oxygen species (ROS) in the activation of NF-kB and the subsequent transcription of more than 200 genes that suppress apoptosis and induce cell transformation, proliferation, invasion, metastasis, chemo-resistance, radio-resistance, and inflammation [49]. In this regard, it has been shown that carvacrol can prevent stress-induced by oxidative damage, decreased levels of malondialdehyde and increased levels of reduced glutathione, and the activity of antioxidant enzymes superoxide dismutase, glutathione peroxidase, glutathione reductase and catalase [31]. Since the antioxidant activity of the EOCc was minor, the antitumor activity is likely due to some other involved mechanisms. Further studies are needed to explain the mechanism of carvacrol and EOCc on breast cancer and its synergic effect with other antitumor drugs.

## 4. Materials and Methods

### 4.1. Plant Material and Carvacrol

5 kg of *C. citratus* leaves were collected in Lima, Botanical Garden of San Fernando, Peru (12°03′28″ S and 77°01′23″ W) on 30 May 2019. Voucher specimen (069-USM-2010) was deposited at the Herbarium, Universidad Nacional Mayor de San Marcos (Lima, Peru). Carvacrol was purchased from the Sigma Chemical Co St Luis, MO, USA.

### 4.2. Extraction of the Essential Oil of Cymbopogon Citratus (EOCc)

The essential oil isolation was carried out by hydro-distillation with a Clevenger-type apparatus for 4 h. The essential oil was dried, and the distillate was separated based on its immiscibility properties and density differences between water and the essential oil, using a glass separation pear. Then, it was dehydrated over anhydrous sodium sulfate, finally filtered, and stored in an amber glass jar under refrigeration (4 °C) until further use [50]. Finally, 0.52% of the essential oil was obtained.

### 4.3. Analysis of the Chemical Composition of the Essential oil of Cymbopogon Citratus (EOCc)

The essential oil of *C. citratus* was analyzed using gas chromatography coupled to mass spectrometry (GC-MS). The analysis was performed in an Agilent Technologies Chromatograph 6890 N series, coupled to a mass spectrometer-detector Agilent series 5973, operated in electron-ionization mode at 70 eV, fitted with a 5% diphenyl and 95% dimethylpolysiloxane capillary column (DB-5 MS, 30 m × 0.25 mm × 0.25 μm). Helium was used as carrier gas (1.00 mL/min in constant flow mode). The injection system operated in split mode (40:1) at 220 °C, with the transfer line at 230 °C. The GC oven temperature was kept at 60 °C, then increased to 250 °C with a gradient rate of 3 °C/min. The ion source temperature was 250 °C. 1 μL of a solution of the oil in CH_2_Cl_2_ (1:100 *v*/*v*) was injected [51]. The oil components were identified through a computer search using the Wiley Registry of Mass Spectral Data (6th edition), and through comparison between calculated linear retention indices and data from the literature [30,52]. Quantitative data were obtained from peak areas using a flame ionization detector (FID). The percentage composition of the oil was determined by correlating GC peak areas to the total chromatogram, without applying any correction factor, but normalizing values with nonane as an internal standard.

### 4.4. DPPH Radical Assay

This protocol was carried out as described by Yu [53] with slight modifications, in 96-well microplates, with a total volume of 200 µL. The test samples were dissolved with methanol and serially diluted at different concentrations (EOCc: 0.5–64 mg/mL, and carvacrol: 6.25–800 µg/mL). 100 µL of each sample dilution was mixed in each well with 100 µL of freshly prepared DPPH solution in methanol (0.2 mM); The control was also performed with 100 µL of methanol, 100 µL of 0.2 mM DPPH, and the blank of the sample with 100 µL of the same dilutions, 100 µL of methanol (without DPPH). Each sample, as well as each control, was analyzed by triplicate. After incubation in the dark for 30 min at room temperature, the microplate was slightly stirred for 5 s and the absorbance was measured at 517 nm with Microplate Reader (Ivdiagnostic., Shenzhen, China). The ability to sequester the DPPH radical was calculated using the following Equation (1):Antioxidant activity (%) = [(A0 − A1)/A0] × 100(1)
where A0 is the absorbance of the control reaction and A1 is the absorbance in the presence of the sample, corrected by the absorbance of the sample itself (blank). The percent inhibition was plotted against the concentration and the half inhibitory concentration (IC_50_) was determined graphically by plotting the absorbance against the used essential oil concentration, calculated by using the slope of the linear regression.

### 4.5. Animals

The research was carried out on 54 female Holztman rats with body weight 160 ± 20 g (age: 6 to 8 weeks). The animals were purchased from Bioterio of the National Institute of Health (National Center of Biological Products, Lima, Peru). Body weights and survival were monitored throughout. Rats had free access to drinking water and basal standard diet ad libitum. Animal experiments conducted in this study conformed to internationally accepted standards and were approved by the Bioethical Committee of the Universidad Nacional Mayor de San Marcos (ID 0272-2018). All experiments were performed according to institutional guidelines in compliance with the requirements of the “European Convention for the protection of vertebrate animals used for experimental and other scientific purposes. [54]

The standard food was prepared by the Universidad Nacional Agraria La Molina. The ingredients were: corn flour, soybean cake, extruded soybean meal, wheat mill by-products, vegetable oil, calcium carbonate, dicalcium phosphate, choline chloride, sodium chloride, amino acids, vitamins, and minerals. The nutritional value was protein (17%), lysine (0.92%), methionine-cysteine (0.98%), fat (6%), calcium (0.63%), phosphorus (0.37%), fiber (4%), humidity (12%) and carbohydrate (58.1%).

### 4.6. Induction of Breast Cancer in Rats and Experimental Design

The induction of breast cancer was performed according to the method of Wang & Shang, 2017 [55]. 7,12-dimethylbenz [a] anthracene (DMBA) was administered at a single dose of 80 mg/kg body weight (BW) per oral gavage, diluted in 1 mL of olive oil. A total of 54 rats were randomly assigned into 9 groups (n = 6). Group I received Physiological saline (control); Group II received DMBA; Groups III, IV and V, DMBA (single dose) + EOCc daily at doses of 50, 100 and 200 mg/kg body weight, respectively; Groups VI, VII and VIII, DMBA (single dose) + carvacrol daily at doses of 50, 100 and 200 mg/kg BW, respectively; and group IX received DMBA (single dose) + EOCc and carvacrol daily at doses of 100 mg/kg/day BW. The essential oil and carvacrol were administered by oral gavage for 14 weeks. A mixture of two solvents to dilute the essential oil such as 1% dimethyl sulfoxide (DMSO) and 2% nonionic surfactant polysorbate 80 (Tween^®^ 80) were used as emulsifying agents.

During the evaluation, the time of appearance of the mammary tumors (latency) was controlled and the body weight of the rats was recorded weekly. At the end of the experiment, the animals were sacrificed by pentobarbital overdose. All tumors were counted in each rat, then removed to determine their volume and histopathological analysis. The cumulative tumor volume was calculated by the formula:(2)$$V=\frac{1}{2}\;\left[\frac{4}{3}\;\pi .a.b.c\right]$$
where *V* = cumulative tumor volume; *a* = width; *b* = length; *c* = height.

### 4.7. Histopathological Analysis of Rat Tumors

A tissue sample from each mammary tumor was fixed at 10% buffered formalin. Then, the tissues were sequentially processed for dehydration and clearance with acetone and xylene. Next, the tissues were impregnated with paraffin, cut by using a microtome, the cuts placed in sheets and stained with hematoxylin and eosin (H&E). Histological evaluation was performed by an optic microscope (BX53. Olympus, Tokyo, Japan).

### 4.8. Statistical Analysis

Data were presented as mean ± standard error. Statistical significance was determined by one-way ANOVA, followed by a post hoc Tukey test. The statistical software SPSS version 19 was used. Values of *p* < 0.05 were considered statistically significant.

### 4.9. Ethical Aspects

In the present investigation, the specifications and recommendations of the Guide for the Care and Use of Laboratory Animals and the correct application of the scope of the Animal Protection and Welfare Law (Law No. 30407) were followed.

## 5. Conclusions

Based on our results, we concluded that the essential oil of *C. citratus* (EOCc) and carvacrol had an antitumor effect independent of the doses. Carvacrol at doses of 100 mg/kg BW exhibited the best effect compared with the other treatments. Carvacrol had a high antioxidant activity compared to EOCc against DPPH radical in vitro. Histopathological findings in breast tissues showed a protective effect of carvacrol at 100 mg/kg BW on DMBA-induced breast cancer in female rats. The main component of the essential oil of *C. citratus* was citral with 72.29% of total content (citral is a mixture of optical isomers-geranial (citral A) and neral (citral B)-which could be mainly responsible for the antitumor effect; otherwise, carvacrol, a fingerprint of *Thymus vulgaris* and *Origanum vulgare*, may be a good drug in which to study its effect in breast cancer progression. The results of our present investigation present experimental evidence for the first time of the EOCc and carvacrol on DMBA-induced breast cancer in female rats.

## Figures and Tables

**Figure 1 molecules-25-03284-f001:**
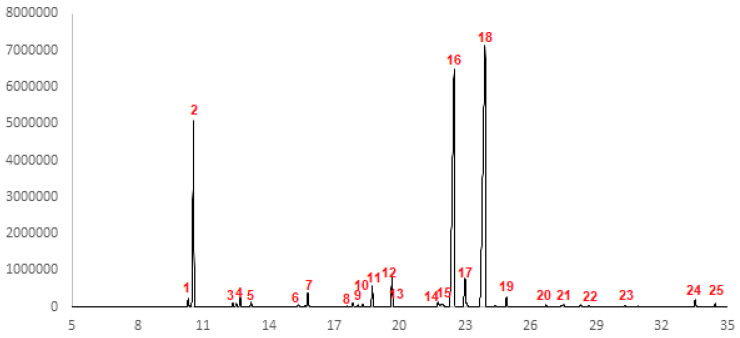
GC-MS Chromatogram for *Cymbopogon citratus* essential oil. Each number corresponds to a chemical constituent found in EOCc detailed in Table 1.

**Figure 2 molecules-25-03284-f002:**
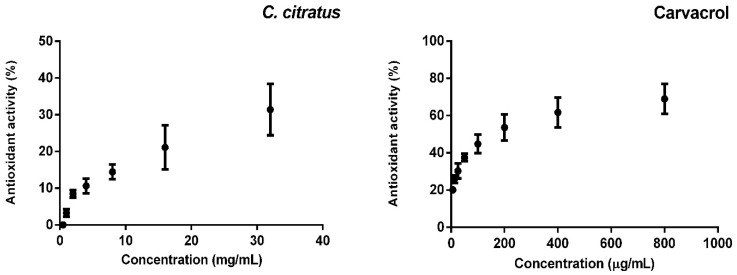
Antioxidant activity by the DPPH of *Cymbopogon citratus* essential oil and carvacrol. All tests were run in three independent experiments (n = 3) and results expressed as mean values with standard deviation (±SD).

**Figure 3 molecules-25-03284-f003:**
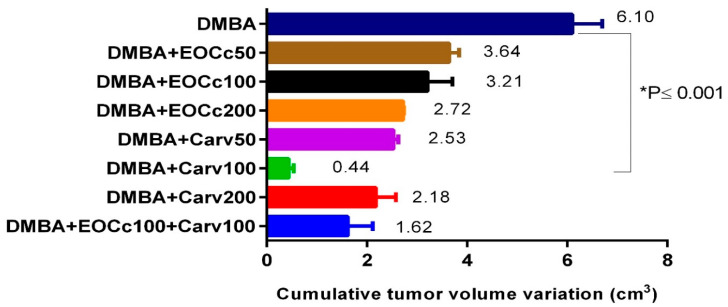
Variation of the cumulative tumor volume as a result of treatment with EOCc and carvacrol in breast carcinogenesis induced by DMBA in rats. * Significant difference from the DMBA group (*p* < 0.001). One-way ANOVA followed by a post hoc Tukey test.

**Figure 4 molecules-25-03284-f004:**
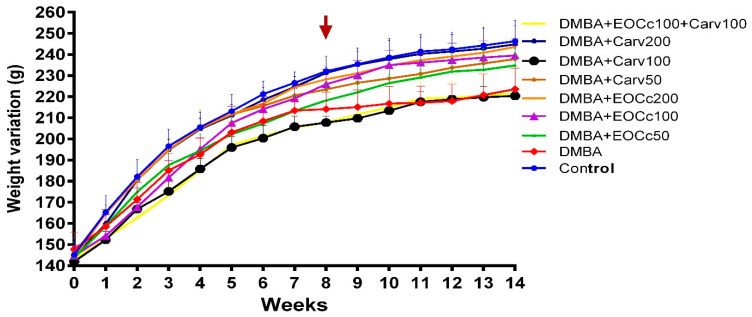
Variation of body weight in rats with DMBA-induced breast cancer and treated with EOCc and carvacrol for 14 weeks. Each point represents ± SEM of six animals. Significant difference from the DMBA group at 8 weeks (*p* < 0.05). One-way ANOVA followed by a post hoc (brown narrow).

**Figure 5 molecules-25-03284-f005:**
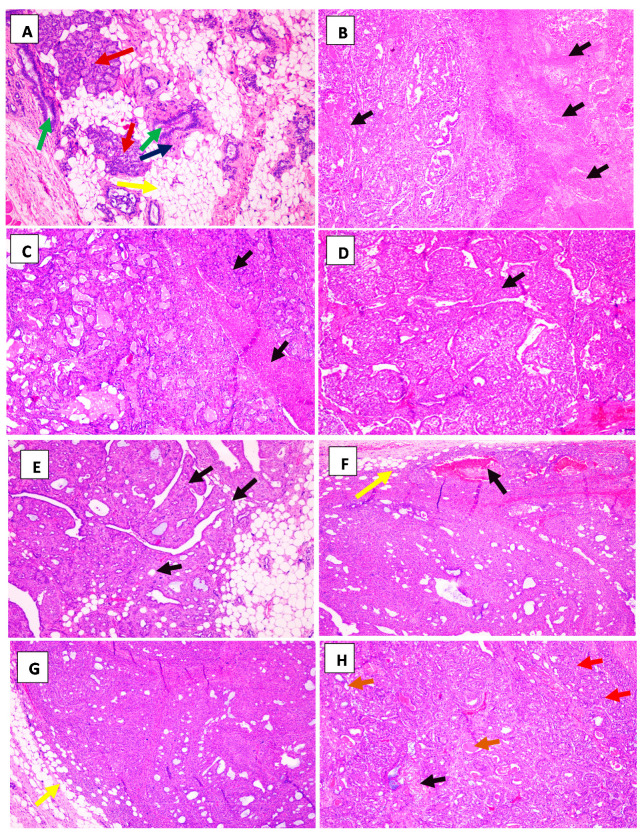
Histopathological profiles (H&E, × 400) of representative mammary tumor tissues from various experimental rat groups (n = 6). (**A**) The normal breast tissue of the control group: mammary acini (red narrows), conduit excretory (green narrows), breast stroma (blue narrow), adipose tissue (yellow narrow). (**B**) DMBA group; infiltrating carcinoma with extensive tumor necrosis (black narrow). (**C**) EOCc-50 mg/kg/day + DMBA; solid-cribriform carcinoma, infiltrating, with necrosis areas (black narrows). (**D**) EOCc-100 mg/kg/day + DMBA, solid infiltrating carcinoma with necrosis area (black narrow). (**E**) EOCc-200 mg/kg/day + DMBA; solid carcinoma with “papillary” areas and infiltrating adipose tissue (black narrows). (**F**) Carvacrol-100 mg/kg/day + DMBA; solid infiltrating carcinoma, necrosis tumor necrosis (black narrow) and infiltrating adipose tissue (yellow narrows). (**G**) Carvacrol-200 mg/kg/day + DMBA; solid carcinoma infiltrating adipose tissue (yellow narrows). (**H**) EOCc 100 + carvacrol 100 mg/kg/day + DMBA; solid carcinoma (red narrows), necrosis (black narrow), tubular areas (brown narrows).

**Table 1 molecules-25-03284-t001:** Chemical composition of the essential oil of *C. citratus* determined by GC-MS.

	Compounds	LRIexp	LRIref	%
1	Not identified	983		0.61
2	Myrcene	988	988	13.60
3	Limonene	1027	1024	0.25
4	1,8-Cineole	1031	1026	0.29
5	(*Z*)-β-Ocimene	1034	1032	0.63
6	(*E*)-β-Ocimene	1045	1044	0.37
7	6,7-Epoxymyrcene	1090	1090	0.22
8	Linalool	1099	1095	1.01
9	exo-Isocitral	1142	1140	0.34
10	trans-α-Necrodol	1147	1144	0.17
11	Citronellal	1151	1148	0.19
12	(*Z*)-Isocitral	1160	1160	1.66
13	(*E*)-Isocitral	1179	1177	2.42
14	Nerol	1224	1227	0.45
15	Citronellol	1227	1223	0.24
16	Neral	1239	1235	31.84
17	Geraniol	1250	1249	2.91
18	Geranial	1270	1264	40.45
19	2-Undecanone	1291	1293	0.73
20	Not identified	1332		0.17
21	Ethyl nerolate	1350	1351	0.25
22	Not identified	1368		0.20
23	(*E*)-Caryophyllene	1415	1417	0.19
24	2-Tridecanone	1493	1495	0.57
25	Myristicin	1515	1517	0.26
Monoterpene hydrocarbons (%)				14.85
Oxygenated monoterpenes (%)				82.17
Sesquiterpene hydrocarbons (%)				0.76
Others (%)				1.23
Total Identified (%)				99.02

LRIexp, Linear Retention Index calculated against n-alkanes C9-C24; LRIref, Linear Retention Index obtained from the literature (Adams 2009) [30].

**Table 2 molecules-25-03284-t002:** Effect of the essential oil of *C. citratus* and carvacrol on DMBA induced breast carcinogenesis in rats.

Parameter/Groups	DMBA	DMBA + EOCc50	DMBA + EOCc100	DMBA + EOCc200	DMBA + CARV-50	DMBA + CARV-100	DMBA + CARV-200	DMBA + EOCc100 CARV-100
Total number of tumors	16.00	11.00	10.00	9.00	9.00	4.00 *	9.00	7.00
Animals with tumors/total of animals	6/6	4/6	5/6	5/6	4/6	2/6	5/6	4/6
Frequency of tumors per group	2.67 ± 0.33	1.83 ± 0.65 (−31%)	1.67 ± 0.42 (−37%)	1.50 ± 0.43 (−44%)	1.50 ± 0.62 (−44%)	0.67 ± 0.49 * (−75%)	1.50 ± 0.43 (−44%)	1.17 ± 0.31 (−56%)
Tumor latency (days)	65.17 ± 2.78	71.00 ± 5.11	75.40 ± 3.12	76.20 ± 3.06	72.50 ± 4.17	79.00 ± 7.00	76.60 ± 3.08	77.25 ± 3.66
Tumor incidence (%)	100.00	66.67 (−33%)	83.33 (−17%)	83.33 (−17%)	66.67 (−33%)	33.33 (−67%)	83.33 (−17%)	66.67 (−33%)
Average tumor volume (cm^3^)	0.38 ± 0.04	0.33 ± 0.02	0.32 ± 0.03	0.30 ± 0.04	0.28 ± 0.03	0.11 ± 0.05 *	0.24 ± 0.03	0.23 ± 0.03
Cumulative tumor volume (cm^3^)	6.10 ± 0.05	3.64 ± 0.10	3.21 ± 0.15	2.72 ± 0.05	2.53 ± 0.08	0.44 ± 0.01 *	2.18 ± 0.02	1.62 ± 0.01 *

Values expressed as mean ± SEM. EOCc; essential oil of *Cymbopogon citratus*, CARV; carvacrol. * Significant difference from the DMBA group (*p* < 0.05). One-way ANOVA followed by a post hoc Tukey test.

**Table 3 molecules-25-03284-t003:** Effect of the essential oil of *C. citratus* and carvacrol on histological parameters in DMBA-induced breast carcinogenesis in rats.

Histological Parameters	DMBA	DMBA + EOCc50	DMBA + EOCc100	DMBA + EOCc200	DMBA + CARV-50	DMBA + CARV-100	DMBA + CARV-200	DMBA + EOCc100 CARV-100
Necrosis (%)	75.0	25.0	20.0	0.0	0.0	0.0	0.0	0.0
Mitosis	1–2	0–1	0–1	0.00	0–1	0	3–5	0–1
Infiltration	positive	positive, muscle	positive, adipose tissue	positive, muscle	positive, adipose tissue	negative	negative	positive, adipose tissue

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
