# Peer review of "The Essential Oil of *Cymbopogon citratus* Stapt and Carvacrol: An Approach of the Antitumor Effect on 7,12-Dimethylbenz-[α]-anthracene (DMBA)-Induced Breast Cancer in Female Rats"

_molecules, 2020, doi:10.3390/molecules25143284_

Round 1

Reviewer 1 Report

Dear authors, although EOC biological activities has already demonstated antitumor in vitro effects, in many studies, the in vivo approach is always mostly desired. For this reason, present paper has a reasonable interest to the reader, in general. Please find below my main comments on your work.

  1. Apart of "chemopreventing effect" , I believe that the title should also contain the "antitumor effect" of tested materials
  2. I did not find a sort review on Cymbopogon citratus phytochemistry (see recent bibliography data e.g. Scientific Reports, 9, article # 8358, 2019, etc)
  3. There is not also a sort comprehensive presentation, on natural products chemopreventing mechanisms (see:  J BUON 15(4), 627-638, 2010, and others)
  4. Results/Discussion sections are usually placed at the end of the text
  5. Discussion seems like another introduction. Present findings are not actually compared in details with previous experimental data
  6. Figures 4, 5 are not clearly understandable (not clear histological signs on necrosis, infiltration, mitosis, relation of body weight/toxicity)
  7. Carvacrol is a phenolic and not terpene constituent not found in EOC. Why Carv is used as a pilot compound, instead of a pure citral (Geraniol/Neral), which seem to exhibit a different mode of action (not cumulative/linear effect on tumors was basically detected)
  8. Could  the degree of antioxidant effect between EOC and Carv be the responsible factor compared, since very different conc. were tested. In this case which are the main mechanisms involved on chemopreventing/anti tumor activities of the EOC (e.g. COX, TNF-a, nitric oxide, caspase etc). A declaration/conclussion is  urgently needed in Discussion (lines 140-148)
  9. The female rats were all in the same  hormone status?
  10. How much plant material was actually extract and which solvent was used? 

Author Response

Dear Reviewer 1:Thank you for your recommendations.

Thank you for your commentaries, we have corrected according to your suggestions:

  1. Apart of "chemopreventing effect”, I believe that the title should also contain the "antitumor effect" of tested materials.

R1:  Thank you dear reviewer, we modified this word.

  1. I did not find a sort review on Cymbopogon citratus phytochemistry (see recent bibliography data e.g. Scientific Reports, 9, article # 8358, 2019, etc)

R2: Line 80-83. We added the main components of essential oils according to recent bibliography.

Brügger, B.P., Martínez, L.C., Plata-Rueda, A. et al. Bioactivity of the Cymbopogon citratus (Poaceae) essential oil and its terpenoid constituents on the predatory bug, Podisus nigrispinus (Heteroptera: Pentatomidae). Sci Rep 9, 8358 (2019).

Major components of C. citratus were Neral (31.5%) and citral (26.1%).

  1. There is not also a sort comprehensive presentation, on natural products chemopreventing mechanisms (see: J BUON 15(4), 627-638, 2010, and others)

R3: Line 90-94. We added according to your suggestions.

  1. Results/Discussion sections are usually placed at the end of the text

R4: Dear reviewer, we written according to template format of the Molecules Journal.

  1. Discussion seems like another introduction. Present findings are not actually compared in details with previous experimental data.

R5. Dear reviewer, we improved our discussion with new references.

  1. Figures 4, 5 are not clearly understandable (not clear histological signs on necrosis, infiltration, mitosis, relation of body weight/toxicity)

R6. Dear reviewer, we improved the figure 4 and 5 with legends and narrows.

  1. Carvacrol is a phenolic and not terpene constituent not found in EOC. Why Carv is used as a pilot compound, instead of a pure citral (Geraniol/Neral), which seem to exhibit a different mode of action (not cumulative/linear effect on tumors was basically detected).

R7: Carvacrol is not found in C. citratus, but it is abundant in other plants, so the objective was to compare its effect with C. citratus essential oil.

  1. Could the degree of antioxidant effect between EOC and Carv be the responsible factor compared, since very different conc. were tested. In this case which are the main mechanisms involved on chemopreventing/anti-tumor activities of the EOC (e.g. COX, TNF-a, nitric oxide, caspase etc).

R8. There is a possibility that the antioxidant effect of C. citrate and carvacrol contributes in part to chemopreventive action. The other tests were not carried out. The chemopreventive effect of C. citrate may be due in part to its high content of citral present in the essential oil. We are going to continue working with these substances in vitro and in silico in order to bioguided our research in cancer.

  1. A declaration/conclusion is urgently needed in Discussion (lines 140-148)

R9.  We did a conclusion section.

  1. The female rats were all in the same hormone status?

R10. All female rats were under the same conditions and probably under the same hormonal status.

  1. How much plant material was actually extract and which solvent was used?

R11. 5 kg of the C. citratus plant was used and 0.52% of essential oil was obtained. We used hydro-distillation. A mixture of two solvents to dilute the essential oil such as 1% dimethyl sulfoxide (DMSO) and 2% nonionic surfactant polysorbate 80 (Tween ® 80) were used as emulsifying agents.

Reviewer 2 Report

In this manuscript, Rojas-Armas et al., studied the chemopreventive effect of EOCc and carvacrol on 7, 12-dimethylbenz [α] anthracene (DMBA)-induced breast cancer in female rat model. The work is potentially interesting, but it is rather descriptive, and almost no mechanistic investigations.

Comments:

  1. “Groups III, IV and V, EOCc daily in doses of 50, 100 and 200 mg/kg body weight, Groups VI, VII and VIII, daily carvacrol in doses of 50, 100 and 200 mg/kg BW, respectively”; What’s the rational to use same dose for both EOCs and carvacrol?
  2. “group IX received both EOCc and carvacrol in doses of 100 mg/kg/day BW,” Explain why did you use such a high dose combination.
  3. The EOCc and citral produced a decreased expression of Bcl-xL and Mcl-1 antiapoptotic factors, Why did not measure these activities in cell culture? You may measure cell cycle progression, apoptosis, signalling pathways.
  4. “the antitumor activity demonstrated in this study due to the effect of carvacrol treatment may be due to its antioxidant activity”. The EOCs and Carvarol have different antioxidant activity. Line 82- The antioxidant activity of the EOCc was slight (IC50 = 48 mg/ml),…… On the other hand, in this study, carvacrol showed antioxidant activity with an IC50 = 145 µg / ml (Figure 2)”. If the doses used in the design was based on their antioxidant activity, Carvacrol should be used 300-fold less.
  5. The best effect was observed with carvacrol at the dose of 100 mg/kg/day. Explain why carvacrol at the dose of 200 mg/kg/day had little protection.
  6. How EOCc and carvacrol were administrated to rats?

Author Response

Dear Reviewer 2: Thank you for your recommendations.

  1. “Groups III, IV and V, EOCc daily in doses of 50, 100 and 200 mg/kg body weight, Groups VI, VII and VIII, daily carvacrol in doses of 50, 100 and 200 mg/kg BW, respectively”; What’s the rational to use same dose for both EOCs and carvacrol?

R1: Dear reviewer, the same doses were used to compare the effect.

  1. “group IX received both EOCc and carvacrol in doses of 100 mg/kg/day BW,” Explain why did you use such a high dose combination.

R2: We used both substances (EOCc + carvacrol) to determine if they showed any synergy.

  1. The EOCc and citral produced a decreased expression of Bcl-xL and Mcl-1 antiapoptotic factors, why did not measure these activities in cell culture? You may measure cell cycle progression, apoptosis, signaling pathways.

R3: Molecular tests are desirable, but require expensive equipment and reagents. We report the results made under the experimental conditions of our laboratory.

  1. “the antitumor activity demonstrated in this study due to the effect of carvacrol treatment may be due to its antioxidant activity”. The EOCs and Carvacrol have different antioxidant activity. Line 82- The antioxidant activity of the EOCc was slight (IC50 = 48 mg/ml),…… On the other hand, in this study, carvacrol showed antioxidant activity with an IC50 = 145 µg / ml (Figure 2)”. If the doses used in the design was based on their antioxidant activity, Carvacrol should be used 300-fold less.

R4: The effect of carvacrol treatment may be due in part to its antioxidant activity; however, in the case of C. citratus, its antioxidant activity was very slight, so its effect is likely due to its main citral component that stops the cell cycle and induces apoptosis (Kapur 2016, reference 43).

  1. The best effect was observed with carvacrol at the dose of 100 mg/kg/day. Explain why carvacrol at the dose of 200 mg/kg/day had little protection.

R5: In many researches the best effect is noted in the minor or middle dose. Carvacrol at doses 100 mg/Kg had better effect than a dose of 200mg/Kg. It works by induction of apoptosis mediated by cell cycle arrest at S phase, increase in Annexin V positive cells, decrease in mitochondrial membrane potential and increase in cytochrome c release from mitochondria, decrease in Bcl2/Bax ratio, increase in caspase activity and cleavage of PARP and fragmentation of DNA. (K.M. Arunasree (2010). Anti-proliferative effects of carvacrol on a human metastatic breast cancer cell line, MDA-MB 231).

  1. How EOCc and carvacrol were administrated to rats?

R6:

R6: Both were administered by oral gavage. We mentioned that in the methodology.

Reviewer 3 Report

The manuscript describes an important information on the anticancer use of essential oil in vivo. However several concerns need to be addressed prior to publications.

(A) Abstract:

  1. a short background on the use of both oils as anticancer particularly against breast cancer will be needed
  2. the description of animal group need to be revised  
  3. Volume up should be changed to volume down

(B) Results:

  1. isolation should be changed to identification
  2. more details should be included to the figure legends including number of replica and types of replica and type of statistical analysis.
  3. very critical piece of data is that combination of both oil showed more number of tumors when compared to carvacrol alone. this is need deep explanation. also why the authors tested only one combination. this should be mentioned carefully in the manuscript including the total concentration and the type of effect produced due to this combination. 
  4. "A significant decrease in body weight gain (P 8 <0.05) was observed in the DMBA group". the amount of reduction should be mentioned in numbers. 
  5. For the histopathological figures, arrows need to be added.
  6. Please check the name of countries  

(C) discussion 

need to be more concise and clear. the authors has so many speculations in relation to his results, although they did not perform any experiments. the authors need to reduce all these speculations and discuss their results in light of previous publications.   

(D) Methods

  1. animal groups to test the effect of each oil alone on the animals are required as control and to test their toxic effects. 
  2. How the IC50 calculated need to be included. 
  3. it is not clear why the author included antioxidant activities in this manuscript. it looks not related.
  4. details regrading the histopathology are needed. 
  5. the authors need to discuss how DMA induce only breast cancers if it is taken orally. more details should be added to introduction and discussion too. 

(E) Although English language looks fine, it still need to be revised carefully and all spelling mistakes need to be corrected including in particular the countries names. 

Author Response

Dear reviewer 3: Thank you for your recommendations in order to improve our article.

The manuscript describes an important information on the anticancer use of essential oil in vivo. However, several concerns need to be addressed prior to publications.

(A) Abstract:

  • a short background on the use of both oils as anticancer particularly against breast cancer will be needed:
  • the description of animal group need to be revised
  • Volume up should be changed to volume down
  • (B) Results:

R1: Dear reviewer, we amended that with a short sentence as background. The description of animal group was reviewed, and changed the word “volume up”.

(B) Results

isolation should be changed to identification

R2: The word was changed.

more details should be included to the figure legends including number of replica and types of replica and type of statistical analysis.

R2: We added some changes to figures (figure 5).

very critical piece of data is that combination of both oil showed more number of tumors when compared to carvacrol alone. this is need deep explanation. also why the authors tested only one combination. this should be mentioned carefully in the manuscript including the total concentration and the type of effect produced due to this combination.

R3: The number of tumors is less with carvacrol alone compared to the combination of carvacrol plus C. citratus essential oil, which shows that the combination does not necessarily increase the effect. However, there is a clear improvement in efficacy compared to C. citratus essential oil in the same dose.

"A significant decrease in body weight gain (P 8 <0.05) was observed in the DMBA group". the amount of reduction should be mentioned in numbers.

R4: The rats' body weight decreased significantly in the DMBA group reaching the average value of 223.67 ± 9.61g compared to 246.33 ± 8.31 g (p <0.05) at the end of the experiment.

For the histopathological figures, arrows need to be added.

R5: We corrected that.

Please check the name of countries 

R6: We corrected the country names, but we included some cities and regions.

(C) discussion

 need to be more concise and clear. the authors have so many speculations in relation to his results, although they did not perform any experiments. the authors need to reduce all these speculations and discuss their results in light of previous publications.  

R7: We reduced some sentences related to mechanisms. However, we are referencing with articles published of carvacrol and essential oil, which its mechanism is detailed in vitro.

(D) Methods

  • animal groups to test the effect of each oil alone on the animals are required as control and to test their toxic effects.

R8:          We did not include another group because essential oil is safety at doses of 2000 mg/Kg according to this article “Phytochemical Analysis and Evaluation of Skin Irritation, Acute and Sub-Acute Toxicity of Cymbopogon citratus Essential Oil in Mice and Rabbits. “ (https://pubmed.ncbi.nlm.nih.gov/31867219/).

Carvacrol is safety up to 2mg/Kg according to “Safety and Tolerability of Carvacrol in Healthy Subjects: A Phase I Clinical Study” Safety and Tolerability of Carvacrol in Healthy Subjects: A Phase I Clinical Study (https://pubmed.ncbi.nlm.nih.gov/30486682/)

  • How the IC50 calculated need to be included.

 R9:  the half inhibitory concentration (IC50) was determined graphically by plotting the absorbance against the used extract concentration or calculated by using the slope of the linear regression.          

it is not clear why the author included antioxidant activities in this manuscript. it looks not related.

              R10: The antioxidant effect is important to evaluate because free radicals are related in mutagenic and carcinogenic processes, thus allowing to partially explain the possible mechanism of anticancer action of the samples tested.

  • details regarding the histopathology are needed.

R11: We corrected figure 5.

  • the authors need to discuss how DMA induce only breast cancers if it is taken orally. more details should be added to introduction and discussion too.

R12: DMBA induces only breast cancer in rats because oxidation of DMBA by CYP 450 enzymes produces metabolites that form DNA covalent adducts and formation within DNA. (Lines 55-61)

 (E) Although English language looks fine, it still need to be revised carefully and all spelling mistakes need to be corrected including in particular the countries names.

 R12: We corrected English grammar.

Reviewer 4 Report

Article ID: molecules-856407-

Title: Cymbopogon citratus Stapt and carvacrol: An approach of the chemopreventive effect on 7, 12-dimethylbenz[α]anthracene (DMBA)-induced breast cancer in female rats

General comments

The article is interesting to the reader, but it needs improvement. There are not enough experimental data, there should be given more preliminary in vitro experiments on cell lines or could be described shortly and cited if Authors did them previously.

Did Authors repeat the experiments from material harvested from other geographical location and month/year? The results could be quite different – similarly to the composition of the oil. It should be discussed.  

The aim of the study is not defined well – is too brief; it should be more descriptive. The research hypotheses should be given.

Detailed comments

  • There is no lines numbering (pp. 1-5) – it hinders the review.
  • References [1] and [2] are too old for the statements in the first paragraph of the Introduction.
  • All abbreviations should be explained, when used for the first time, e.g. “HER2”.
  • Why Authors chose carvacrol for their research? Justify.
  • The main components of the achieved EO are myrcene and geranial (citral). Maybe Authors should make the research in comparison to these two compounds.
  • Why animals were fed for 14 weeks? Justify.
  • Paragraph 2.3.3. The photos are very nice, but Authors should describe first the typical control and then compare the differences after exposition. The description is too brief.
  • Figure 5. Give the scale; what is the magnification? The changes should be marked with e.g. different colour arrows.
  • Lines 137-141. The conclusions are too far, as Authors did not make such experiments… Authors could make ROS generation.
  • Authors should explain/mark that geranial is citral? In Table 1 it should be marked, as Authors say in line 145 “In this study, it has been shown that citral, the main component of the EOCc…”
  • What was the diet of the animals? Give some information about the feed.
  • Materials and Methods description should be improved. Give more detail.
  • Paragraph 4.7. Give more details. Give the information about the microscope, magnification, camera and software.
  • I can’t see specified Conclusions.

Author Response

Dear Reviewer 4: Thank you for your recommendations in order to improve our manuscript.

Title: Cymbopogon citratus Stapt and carvacrol: An approach of the chemopreventive effect on 7, 12-dimethylbenz[α]anthracene (DMBA)-induced breast cancer in female rats

General comments

The article is interesting to the reader, but it needs improvement. There are not enough experimental data, there should be given more preliminary in vitro experiments on cell lines or could be described shortly and cited if Authors did them previously.

R1: Thank you dear reviewer, we did not do any experiment in vitro, but we found several literatures which were cited with works in vitro of the essential oil and carvacrol on breast tumor cell lines as well as their main mechanisms.

Did Authors repeat the experiments from material harvested from other geographical location and month/year? The results could be quite different – similarly to the composition of the oil. It should be discussed. 

R2: Thank you dear reviewer, we did not repeat any experiment from material harvested from other regions, we are aware that its chemical constituents could vary, however, we specify in the method section how C. citratus was collected and stablished conditions.

The aim of the study is not defined well – is too brief; it should be more descriptive. The research hypotheses should be given.

R3: The research hypothesis is that C. citratus essential oil and carvacrol have a chemopreventive effect on DMBA-induced breast cancer in rats.

Detailed comments

There is no lines numbering (pp. 1-5) – it hinders the review.

R4: It was corrected.

References [1] and [2] are too old for the statements in the first paragraph of the Introduction.

R5: It was corrected with new references.

All abbreviations should be explained, when used for the first time, e.g. “HER2”.

R6: It was corrected.

Why Authors chose carvacrol for their research? Justify.

R7. Carvacrol was chosen for research because there is a history of the in vitro effect on the metastatic human breast cancer line for the essential oil and carvacrol.

The main components of the achieved EO are myrcene and geranial (citral). Maybe Authors should make the research in comparison to these two compounds.

R8: Dear reviewer, Comparison can be made with myrcene and geranial, but we chose carvacrol because it is not present in C. citratus and explore the effect of a compound present in other medicinal plants.

Why animals were fed for 14 weeks? Justify.

R9: Dear reviewer, our experimental model used for the study includes this period of time.

Paragraph 2.3.3. The photos are very nice, but Authors should describe first the typical control and then compare the differences after exposition. The description is too brief.

R10: Dear reviewer, we corrected that in figure 5.

Figure 5. Give the scale; what is the magnification? The changes should be marked with e.g. different colour arrows.

R11: Dear reviewer, we corrected that in figure 5 with narrows of different color.

Lines 137-141. The conclusions are too far, as Authors did not make such experiments… Authors could make ROS generation.

R12: Dear reviewer, we agree to do ROS generation but we are working in another project in vitro and vivo as well as in silico study.

Authors should explain/mark that geranial is citral? In Table 1 it should be marked, as Authors say in line 145 “In this study, it has been shown that citral, the main component of the EOCc…”

R12: Dear reviewer, Citral is a mixture of optical isomers; the geranial (citral A) and neral (citral B).

What was the diet of the animals? Give some information about the feed.

R13: Dear reviewer, the standard rat food was prepared by the Universidad Nacional Agraria La Molina. The ingredients were: corn flour, soybean cake, extruded soybean meal, wheat mill by-products, vegetable oil, calcium carbonate, dicalcium phosphate, choline chloride, sodium chloride, amino acids, vitamins and minerals. The nutritional value was: protein (17%), lysine (0.92%), methionine-cysteine ​​ (0.98%), fat ((6%), calcium (0.63%), phosphorus (0.37%), fiber (4%), humidity (12%) and carbohydrate (58.1%).

Materials and Methods description should be improved. Give more detail.

R15: Thank you dear, it was improved and modified.

Paragraph 4.7. Give more details. Give the information about the microscope, magnification, camera and software.

R16: Dear reviewer, An optic microscope (BX53. Olympus, Tokyo, Japan) was used. Magnification was 400X and included in figure 5 legend.

I can’t see specified Conclusions.

R17: I t was included in the conclusion section.

Round 2

Reviewer 1 Report

Revised version has mostly responded to recommendations made, in the first place, resulting to a significant overall article improvement.

Author Response

Dear reviewer, thank you in advance.

Reviewer 2 Report

I recommend to accept this revised version for publication.

Author Response

Dear reviewer, thank you in advance.

Reviewer 3 Report

Although the authors have made lots of modifications, some minors need to be addressed including

  1. the animal groups in the abstract need to be described with inclusion of which group received DMBA.
  2. More details need to be added to all figure legends. figure legend for figure 2 is missing 
  3. all the author answer to address my previous comments need to be included in the body of the text and not as an answer.
  4. the authors did not change the discussion, which I still believe need to be focused and concise 

Author Response

Dear reviewer:

Thank you for your suggestions:All modifications were highlighted with pink color.

  1. the animal groups in the abstract need to be described with inclusion of which group received DMBA. R1:It was corrected 
  2. More details need to be added to all figure legends. figure legend for figure 2 is missing. R2: Figure 2 and other figures were added their legends.
  3. all the author answer to address my previous comments need to be included in the body of the text and not as an answer. R3: Thank you dear reviewer, it was corrected and changes were highlighted with pink color.
  4. the authors did not change the discussion, which I still believe need to be focused and concise. R4: Thank you dear reviewer, the discussion was modified.

Reviewer 4 Report

 Accept in present form.

Author Response

Dear reviewer, thank you in advance.